SOIL Discuss., doi:10.5194/soil-2015-82, 2016 Manuscript under review for journal SOIL Published: 15 January 2016 © Author(s) 2016. CC-BY 3.0 License.

This discussion paper is/has been under review for the journal SOIL. Please refer to the corresponding final paper in SOIL if available.

# Natural versus anthropogenic genesis of mardels (closed depressions) on the Gutland plateau (Luxembourg); archaeometrical and palynological evidence of Roman clay excavation from mardels

J. M. van Mourik<sup>1</sup>, D. J. G. Braekmans<sup>2,4</sup>, M. Doorenbosch<sup>2</sup>, W. J. Kuijper<sup>2</sup>, and J. van der  $Plicht^{2,3}$ 

<sup>1</sup>Institute for Biodiversity and Ecosystem Dynamics (IBED), University of Amsterdam,

Science Park 904, Amsterdam, the Netherlands

<sup>2</sup>Faculty of Archaeology, University of Leiden, Einsteinweg 2, 2333CC Leiden, the Netherlands

<sup>3</sup>Centre for Isotope Research (CIO), University of Groningen, Nijenborgh 4,

9747AG Groningen, the Netherlands

<sup>4</sup>Materials Science and Engineering, Delft University of Technology, Mekelweg 2, 2628CD Delft, the Netherlands

Received: 14 November 2015 – Accepted: 4 December 2015 – Published: 15 January 2016 Correspondence to: J. M. van Mourik (j.m.vanmourik@uva.nl)

Published by Copernicus Publications on behalf of the European Geosciences Union.

#### Abstract

Mardels, small closed depressions, are distinctive landforms on the Luxembourger Gutland plateau. In the present landscape most mardels are shallow fens, filled with colluvial sediments. The genesis of mardels has been studied intensively, inside and out-

side Luxembourg. Some researchers suggested a natural development and consider mardels as subsidence basins due to subsurface solution of gypsum veins, other researchers suggested cultural causes and consider mardels as prehistorical quarries.

In the Gutland, mardels occur on various substrates. Mardels on the Strassen marls (li<sup>3</sup>) are abandoned quarries, related to clay excavation in Roman Time. Mardels on the Luxembourger sandstone (li<sup>2</sup>) are sinkholes, related to joint patterns in the sandstone formation. Mardels on the Keuper marls (km<sup>1,3</sup>) are originally subsidence basins, related to subsurface dissolutions of gypsum lenses and veins, filled with colluvial clay. The results of pollen analysis and archaeometrical tests demonstrate Roman extraction of clay for the production of ancient ceramics. So, the natural depressions have been enlarged to the present mardels. After excavation, the sedimentation of colluvium restarted in the abandoned quarries.

#### 1 Introduction

The geological structure (Fig. 1) of the Luxembourger Gutland is inherited from Tertiary and Quaternary landscape evolution, which was initiated by the Pliocene tectonic uplift of the entire region Luxembourg (Lucius, 1948). The present Gutland is a cuesta landscape, underlain by alternating tilted sedimentary rock formations with different resistance to weathering and erosion.

During the Pleistocene the landscape has been subjected to several cycles of weathering and erosion by the alternation of glacial and interglacial periods (Lucius, 1948; Verhoef, 1966). During glacial periods landscape development was dominated by ero-