# Peer review of "Natural versus anthropogenic genesis of mardels (closed depressions) on the Gutland plateau (Luxembourg); archaeometrical and palynological evidence of Roman clay excavation from mardels"

_SOIL, 2015_

## Referee Comment (RC1) · G. Huckleberry (Referee) · 25 Jan 2016

Synopsis

Authors try to demonstrate an anthropogenic origin for several closed depressions (mardels) on the Luxembourg Gutland Plateau that are formed in sedimentary rock. Specifically, they propose that some of these mardels are the vestiges of Roman quarries that were used for clay extraction, primarily in the production of ceramics. Data

set includes sediments collected by bucket auger and a peat core for pollen analysis from fens within three newly studied mardels (Medernach, Brasert2, and Michelbouch) and one mardel (Dauwelsmuer) whose pollen sequence was previously documented by Schwenninger (1989) ; new macrobotanical analysis is also performed for Dauwelsmuer. The pollen data are supplemented by particle size, pH, and geochemical analysis (XRF) analysis on five sediment samples from the mardels, five sediment samples from nearby soils, and four ceramics obtained from nearby Michelbouch. Chronometry for the deposits filling the mardels is based solely on pollen evidence except for one (Dauwelsmuer) whose fill was previously dated with 14C (peat deposits) by Schwenninger. If I'm reading this correctly, the ages for the other mardels are correlative ages where pollen spectra are correlated to the dated sequence at Daulwelsmuer (Table 1). The authors provide a Holocene paleoenvironmental reconstruction for the area and argue that the fills post-date Roman occupation and have a similar clay mineralogy to the analyzed ceramics. They conclude that some of these mardels are natural depressions formed through dissolution and collapse of underlying substrates whereas others are indeed Roman clay quarries.

General Comments

The main contribution of this study is the analysis of pollen from fens formed in these mardels that provide insight into local/regional Holocene vegetation changes due to climate and/or land use changes. The chronology is not very robust, limited to correlation to a previously dated pollen sequence at Daulwelsmuer, but at a minimum provides evidence for post-glacial environmental changes through time.

Unfortunately, the paper is poorly written and difficult to follow in places; it requires heavy copy editing. However, the main problem is that insufficient data are presented to support the conclusion of an anthropogenic origin for any of the mardels. Not enough contextual information is provided to explain the sampling strategy and location of the various samples in relation to one another. There are no large scale maps of the individual mardels showing size, depth, etc.; no locations of sediment cores within the

mardels; no locations of the soil and ceramic samples in relation to the mardels; and no stratigraphic columns with detailed sedimentological/pedogenic descriptions of the mardel fill deposits (this is especially problematic given that the paper is submitted to SOILD). There is virtually no information on the origin and nature of the ceramics (e.g., find locations, thin sections, stylistic features, etc.).

Numerical ages based on palynological correlation are not well explained. More information is needed on the palynochronological markers common to this region, preferably in the background section prior to Results and Discussion. And despite claimed problems with 14C dating organic matter in mardels (p. 8), the 14C chronology from Daulwelsmuer is claimed to be "reliable". How do we know that? Much rests on the accuracy of the age estimates but little is presented explaining the chronology and stratigraphy of the reference mardel.

Another problem is that results and interpretations are mixed throughout the document. Results should review the pollen spectra of the four mardels, the correlated ages, and the archaeometric results. Later, in the Discussion (or Interpretation) section, inferences can be made regarding climate, land use, and vegetation changes on the plateau, and possible provenance of the clay used in the Roman ceramics.

Unfortunately, the weakest part of the study is the "archaeometric" analysis which seems like an afterthought. One sediment sample from each mardel is unlikely to be representative......likewise, with the adjacent soils. The limited sample size and contextual information for mardel fill, soils, and ceramics precludes the ability to determine provenance with the given geochemical results and statistical analyses. Without better definition of the sample universe, it is not possible to determine a reliable match.

Finally, given that the provenance of the clay in the Roman ceramics is not demonstrated, and that the chronology of the newly studied mardel deposits is tenuous, the conclusion that some of the mardels were used by Romans as clay sources is not supported by the data at hand.
Due to these concerns, I cannot recommend that this paper be published, even with major revisions. The authors need to review the international literature on ceramic provenance studies and think about how best to devise a robust sampling strategy that will result in an adequate data set that can support interpretations of Roman clay quarrying in these interesting landscape features. Such a study might have a better chance of being published in an archaeometry journal.

---

## Referee Comment (RC2) · S. Shelley (Referee) · 12 Feb 2016

I am not going to provide a summary of the paper, Gary Huckleberry provides a fairly clear and concise summary in his comments, and the authors provide an abstract. I do want to applaud the authors for applying multiple lines of evidence from several disciplines to address the issue. I think that this makes this paper appropriate for this journal.
[Figure]

General Comments

I did find the paper to be a bit difficult to follow in places. Crucial pieces of information are just thrown out without adequate discussion of why they will be important. For example the four spikes in the Fagus pollen in Dauwelsmauer (F1-F4) are presented with dates on Page 11 Line 20 as part of the description of the pollen sequence. Later during the discussion of each of the other mardels these spikes are simply referred to as F1, F2, etc. These will come back later to play an important role in the discussion of the filling and pollen sequence in the mardels, and used as one of the primary dating methods. I found this frustrating and confusing since initially I wasn't sure what these designations meant; I had to spend a fair amount of time looking back over the paper to find them.

I think in the interest of people that might be unfamiliar with the Roman occupation of the region a quick summary is in order. This area was brought under Roman rule by Julius Caesar in 53 B.C during his conquest of greater Gaul. Initially it would have been little changed by the Romans, but as the region was assimilated into the Roman Empire it took on an increasingly Roman character – Roads, trade, villas, and commercial enterprises. This should have included pottery production, but it should also have included the production one of the most ubiquitous of Roman building materials – brick. As the Franks began to push into the region, by then the Roman province of Gallia Belgica, they forced the Romans to abandon the region by A.D. 406. If the mardels were quarried by the Romans, 53 B.C. to A.D. 406 is the time frame of interest for the anthropogenic origins by the Romans. If the authors are going to try to make the case that the Romans quarried the mardels they need to be more explicit on the time frame.

The paper lacks explicit assumptions and needs to be more explicit in its conclusions. It was fairly clear to me that the authors are relying not only on the dating of Daulwelsmuer but also cross-correlating their results with what is generally known about the climate and palynology of Western Europe, but I don't recall that this is ever explicitly stated as such. The same could be said about the significance of the trun-

cated paleosols. A truncated paleosol is a discontinuity in the soil sequence and is a result of either erosion or the Romans having removed that material. I would like to see more discussion on this discontinuity since I think the anthropogenic argument hinges on this. The authors are at least explicit in stating the overlying sediments and subsequent soil development is post-Roman, and I think they are probably correct in their assertions.

Huckelberry's discussion of the dating issues is fairly thorough and I agree with most of his comments. I do think that cross-dating using the pollen diagrams is acceptable, however it is a weak case argument because it is not very precise, and there is no control over the rates of sedimentation. In the Daulwelsmuer pollen diagram it is difficult to determine just where the Roman occupation would fit in the diagram. The dates go from 4260 BP (about 1700 years before the start of the Roman Republic) to 555 BP (1000 years after the Romans abandoned Gallia Belgica). I think the authors are using the F1 spike in Fagus pollen to mark the Roman period, but they aren't specific about using that as their marker. This brings me to another point. There is a long discussion of the pollen diagrams and how they relate to the climatic sequence and palynological indicators. I thought it was fairly well presented, documented and referenced, and not quite to the point. The article is about whether mardels are anthropogenic, specifically Roman, or natural phenomenon, not the climatic sequence of Europe and its effects on the environment. Discussions of the pollen from the Little Ice Age or the early Holocene don't appear to be particularly relevant, other than to provide confidence in the dating by cross-referencing pollen diagrams. I think the discussion of the pollen could be tightened up and again made more explicit about how it applies to the topic at hand.

As for the archaeometric analysis it doesn't seem like an afterthought to me, I think it is an important piece of the research. As Huckelberry correctly points out it is hampered by a small sample size, it can suggest conclusions but they are not well supported. I think their case can be made without detailed sediment descriptions and information

on precisely where samples were collected, however, such information would greatly strengthen their results. The archaeometric study says it uses Roman ceramics. As a North American archaeologist that means pottery to me, but it could mean ceramic tiles, bricks or other items in this context, the authors should be specific as to what types of ceramic. My guess is that it is pottery. At least this analysis produced an explicit conclusion (page 20, lines 18-20) that Beaufort and Berdorf are not potential sources for local ceramics. Again, a more representative sample size would increase confidence in this result.

If I could make a suggestion – if the authors want to pursue this line of evidence in the future, and I think they should, try using Roman bricks, if available, rather than pottery for testing. Pottery tends to be traded and can move very long distances for many reasons. Bricks, on the other hand, are bulky, heavy items that are economically unfeasible to move long distances (except possibly by ship). The Romans used brick so ubiquitously that there must be Roman brick buildings near these mardels. The brick from these buildings are less likely to be imported and more likely locally produced than the ceramics, and if the mardels are used to produce ceramics, then there is a high probability that it would be bricks.

I am going to conclude by saying I liked the approach to this problem and I think it meets the scientific significance of the journal. I think the scientific quality is mixed – the understanding of the methods and documentation and referencing is good to excellent, but the discussion tends to be unfocused with respect to the stated problem and sample size and dating issues make the arguments weak and poorly supported. The presentation quality needs work, I would say it was fair, it would stand with a major revision (see Huckleberry's comments). I read this paper and made my notes prior to looking at Huckelberry's comments. The two of us independently had many of the same issues, I refer to Huckleberry's comments where we agreed rather than reiterate them. I disagree with Huckleberry that the paper should not be published, I think the authors should be allowed a chance to revise their presentation. However, I think even

if they do rewrite the paper, their conclusions will still be tentative because I don't see
that they can address the issues of sample size and dating without more research.

---

## Referee Comment (RC3) · J. Onken (Referee) · 29 Feb 2016

General comments:

Van Mourik and co-authors investigate the origin of mardels (closed depressions) in Luxembourg. They date the deposits with palynological correlation and markers and use soil and ceramic XRF data to infer that the Roman-period occupants of Bois Biischtert (and likely elsewhere) enhanced pre-existing mardels by clay quarrying. I found

the premise of this study and its objectives interesting. The paper itself, however, is poorly written and structured and this made it challenging to evaluate the research's merits and flaws. Because of this, I would argue that the manuscript fails to meet the principal evaluation criteria of SOIL. As such, I recommend major revision followed by re-evaluation by the topical editor and, if warranted, a second round of peer review.

The below list of specific comments is not comprehensive, but includes numerous, representative problems or concerns. After compiling my notes, I read the reviews submitted by referees Huckleberry and Shelley. Both of these reviews bring up valid points and make many constructive suggestions. Shelley's comments regarding the omission of background on Roman archaeology and the need to describe the ceramics studied are important. Although the authors state (p. 21, line 13) that their ceramics are potsherds (i.e., not bricks, as Shelley was wondering), the fact that they are analyzing pottery should be stated much earlier in the paper. I have to agree with Huckleberry that the data–at least as presented–may not sufficiently support the conclusions. I wonder if this work might be better suited to a publication format geared toward preliminary findings and research in progress.

Specific Comments:

1. Consider a more succinct title, preferably without a colon or semi-colon.

2. In the abstract, intro, and perhaps title, mention that the Luxembourger Gutland is in western Europe. SOIL is an international journal and you should not assume your readers have geographic familiarity with the area or with Roman archaeology. Also, revise to make clear what info in the abstract is background and what info is your findings. Add a sentence or two to the abstract summarizing the implications and importance of your findings.

3. Overall, the structural organization of the manuscript needs revamping. Many of the paragraphs are poorly organized, consisting of disjointed, choppy, poorly connected sentences. In places, the prose is convoluted and gaps in logic were evident. The

overuse of 1–2 sentence paragraphs contributes to this. The excessive use of bulleted lists in the discussion section is distracting.

4. Some issues seem to stem from issues with English fluency and/or translation.

5. The abbreviation CD is not defined. I assume this is "closed depressions"? Use of this abbreviation probably isn't warranted.

6. The number of figures (14) is probably excessive and not justified. Perhaps some should be culled or provided as supplementary information.

7. Why aren't geologic map unit abbreviations (e.g., li2, km1, etc.) indicated on Figure 1? If you don't use them on the figure, they probably should be removed from the text.

8. Although the pollen-based dating of your cores is not particularly robust, it seems reasonable. That said, your inference of a post-Roman age for the mardels would be stronger if confirmed by a few radiocarbon dates from mardel deposits other than at Dauwelsmuer. Is all organic material in your study mardel deposits truly compromised by reservoir effects? Why didn't you do macro analysis on all the mardel study sites, as this might have resulted in 14C-dateable material as it did at Dauwelsmuer? Alternately, could you have radiocarbon-dated pollen or used optically simulated luminescence (OSL) dating?

9. Mardel Dauwelsmuer location should be shown on Figure 1.

10. P. 10, line 24, reference should be to Table 1, not Table 2.

11. What is the significance of the colors used in Figure 4?

12. In Section 3.1, I question your usage of the term "formation". It seems like "stratum" might be more appropriate in this context.

13. In Table 2, the soil texture percentages do not add up to 100%. Why is only one soil sample per profile analyzed? What horizons were analyzed? Why aren't representative soil profile descriptions provided?

14. Are the four ceramic sherds you analyzed truly representative of the full range of Roman pottery at site Bois Biischtert? How did you select this sample? More justification of your sampling strategy is needed, especially because of the small number sampled.

15. Some methods are mixed in with the results, such as p. 15 lines 26-27 and p. 16 lines 1-2.

16. The graph axes are erroneously labeled on Figure 11.

17. P. 17, line 17: do you mean synsedimentary? If so, include definition to clarify.

18. In your introduction, clarify whether any previous studies have used similar methods to address mardel genesis (it appears they haven't, but you should state this explicitly). Also, have prior Roman pottery sourcing studies been done? If so, what were the relevant findings?

19. The discussion included in section 4.1 fails to clearly demonstrate an anthropogenic origin. You are implying that the age of the mardel deposits is consistent with an anthropogenic origin, but correlation does not necessarily indicate causation. I think my confusion here indicates minimally that your logic and conclusions in this section are not clear or explicit. Alternatively, perhaps part of the problem is the first paragraph of section 4.1–I wonder if it should be moved to the beginning of section 4.3?

20. Section 4.2 states, "The results show a reliable matching of the properties of the Roman potsherds...with the clay samples..." This section lacks an adequate discussion of the evidence for this conclusion. It also lacks reference to relevant figures/tables.

21. You conclude that the archaeometric (XRF) similarity of the Roman sherds with the clay samples from the Steinmegelkeuper mardels and surrounding soils indicates that the mardels have an anthropogenic origin. This argument as presented seems weak.

Couldn't clay have been collected on a smaller scale from small, scattered soil pits without the mardels necessarily having a human origin? Is there other corroborating evidence that supports your inference? Please clarify and strengthen your argument for this conclusion.

22. What are the implications and importance of your findings? These belong in the conclusions (and perhaps the discussion), as well as the abstract.

---

## Author Comment (AC1) · 16 Apr 2016

Natural versus anthropogenic genesis of mardels (closed depressions) on the Gutland plateau (Luxembourg); archaeometrical and palynological evidence of Roman clay excavation from mardels J. M. van Mourik, D. J. G. Braekmans, M. Doorenbosch, W. J. Kuijper and J. van der Plicht.

We thank the reviewers (Huckleberry, Onken and Shelly) fort their carefully reviews and

the critical comments. and detailed remarks (Shelley) that we can use to improve the quality of this paper improve our paper.

This paper aims to contribute to the questions about the genesis of closed depression (mardels) on the Gutland plateau in Luxembourg (NW-Europe). Previous research results (1963-1999) indicated dissolution of gypsum lenses, occurring in the Keuper marls as responsible. In more recent publications (2011-2015) the CD's were explained as (Roman) clay excavations based on the observation that the (present) mardel fillings were dated as Post Roman. These researchers did not pay (any) attention to the soil processes, responsible for soil erosion and colluvial deposition in mardels on the plateau.

We tried to contribute to the mardel discussion by following a sequence of research steps. 1. Description of the controlling soil processes, responsible for colluvial deposition in mardels (development of stagic Alisols, lateral clay leaching). On the gentle slopes of the plateau, mardels are sediment traps; mardel sediments have a higher clay content then the soils in the surrounding. 2. Dating of the colluvial deposits by (pollen analysis, including pollen density). Special attention was paid on the (palynological) properties of the boundary between mardel deposits and the underlying (paleo)sol. 3. For the interpretation of the pollen diagrams of mardel deposits, we had to create a reliable schedule of the palynological time markers because dated palynological references for Luxembourg are hardly available. For that reason we resampled Dauwelsmuer. Previous investigated and published by Schwenniger, but the vertical resolution of that diagram was insufficient and radiocarbon dates were not available. The present diagram has a much better resolution and a perfect registration of the relevant palynological time markers, sustained by reliable radiocarbon dates; the horizontal distance to the Strassen marls mardels is less than 10 km, to the Keuper mardels less than 20 km. 4. Close to a Keuper mardel cluster, east of Michelbouch was an archeological Roman excavation spot(Biischtert). This provided us the unique possibility to test the probability if Romans really used mardel clay for the production of pottery. Of

course we tried to find more samples of pottery, but on the Gutland plateau was in 2015 the only available spot.

We agree with the reviewers that the structure of this paper can be improved by following strictly the four mentioned steps. Dauwelsmuer (location must be added in fig.1) is not a mardel on the Gutland plateau (altitude ≈ 400 m) but a small basin in in the debris of a landslide on a sandstone escarpment (altitude 244 m). The controlling soil process in weathered sandstone is tends to podzolizing, not clay leaching. In this depression pure peat could accumulated from ≈6500 BP till ≈550 BP. Purified leaf particles were used for reliable radiocarbon dating. In the next version of this paper we will use the pollen diagram (fig.4) just for a correct description of the palynological time markers that we can use to date the mardel sediments. Also the period that Romans were present in the region needs more attention. In contrast to the study of Schwenninger, our aim is not a description of the local vegetation development, so this part will be skipped. Organic matter in mardel sediments are not reliable for radiocarbon dating because of the accumulation of (upslope) eroded soil organic matter. This concerns all humic fractions, including (very scares) macro remains. An exception was the thin peat layer that we found in the Beaufort4 mardel (one of a cluster of five mardels on the Strassen marls) between the quarry floor and the colluvial strata. We used this peat for reliable radiocarbon dating. OSL dating cannot be applied on clayey sediment. What about the amount of samples, used for the XRF analyses, we will specify better the sample size and add more data to the set. From the Biischtert spot we have pottery samples as well as brick stone. However, we have to accept that this is the only available spot for samples. We used this unique possibility for the XRT analysis to try to confirm the anthropogenic origin of mardels on the marls.